# An Aminopyrimidone and Aminoimidazoles Alkaloids from the Rodrigues Calcareous Marine Sponge *Ernsta naturalis*

**DOI:** 10.3390/md20100637

**Published:** 2022-10-13

**Authors:** Pierre-Eric Campos, Gaëtan Herbette, Laetitia Fougère, Patricia Clerc, Florent Tintillier, Nicole J. de Voogd, Géraldine Le Goff, Jamal Ouazzani, Anne Gauvin-Bialecki

**Affiliations:** 1Laboratoire de Chimie et de Biotechnologie des Produits Naturels, Faculté des Sciences et Technologies, Université de La Réunion, 15 Avenue René Cassin, CS 92003, CEDEX 9, 97744 Saint-Denis, France; 2Institut de Chimie Organique et Analytique, Université d’Orléans—CNRS—Pôle de Chimie, Rue de Chartres—UMR 6759, BP6759, CEDEX 2, 45067 Orléans, France; 3CNRS, Aix-Marseille Université, Centrale Marseille, FSCM, Spectropole, Campus de St Jérôme-Service 511, 13397 Marseille, France; 4Naturalis Biodiversity Center, Darwinweg 2, 2333 CR Leiden, The Netherlands; 5Institute of Environmental Sciences, Leiden University, Einsteinweg 2, 2333 CC Leiden, The Netherlands; 6CNRS, Institut de Chimie des Substances Naturelles, UPR 2301, Université Paris-Sud, Université Paris-Saclay, 1, Av. de la Terrasse, 91198 Gif-sur-Yvette, France

**Keywords:** *Ernsta naturalis*, marine sponge, aminoimidazolones alkaloids, aminopyrimidone alkaloid, molecular network

## Abstract

A chemical study of the CH_2_Cl_2_−MeOH (1:1) extract from the sponge *Ernsta naturalis* collected in Rodrigues (Mauritius) based on a molecular networking dereplication strategy highlighted one novel aminopyrimidone alkaloid compound, ernstine A (**1**), seven new aminoimidazole alkaloid compounds, phorbatopsins D–E (**2**, **3**), calcaridine C (**4**), naamines H–I (**5**, **7**), naamidines J–K (**6**, **8**), along with the known thymidine (**9**). Their structures were established by spectroscopic analysis (1D and 2D NMR spectra and HRESIMS data). To improve the investigation of this unstudied calcareous marine sponge, a metabolomic study by molecular networking was conducted. The isolated molecules are distributed in two clusters of interest. Naamine and naamidine derivatives are grouped together with ernstine in the first cluster of twenty-three molecules. Phorbatopsin derivatives and calcaridine C are grouped together in a cluster of twenty-one molecules. Interpretation of the MS/MS spectra of other compounds of these clusters with structural features close to the isolated ones allowed us to propose a structural hypothesis for 16 compounds, 5 known and 11 potentially new.

## 1. Introduction

Calcispongiae (Calcarea Bowerbank, 1864), commonly called calcareous sponges are much less studied chemically compared to another class of Porifera, the Demospongiae, due both to their relatively low number of representatives within the Porifera phylum and to their low biomass [1]. The result is an underexplored source of natural products while these sponges are prolific sources of bioactive alkaloids, especially 2-aminoimidazole alkaloids. The biological activities reported for this kind of alkaloids include antifungal [2], antimicrobial [3], cancer cell toxicity [4], or Mammalian and Protozoan DYRK and CLK kinases inhibitors [5]. Nowadays, more than sixty 2-aminoimidazole alkaloids have been isolated from Calcarea sponges, almost all belonging to the genus *Leucetta*. A few studies described the chemical composition of other calcareous sponges’ genera such as *Clathrina* [6], *Leucosolenia* [7], *Leucascandra* [8,9], or *Pericharax* [10,11]. The genus *Ernsta* (Klautau, Azevedo and Cóndor-Luján, 2021) belongs to the order Clathrinida, and comprises 20 species, and despite a ubiquitous distribution, there is no report of chemical investigations of sponges belonging to this genus so far. The genus *Ernstia* was erected by Klautau et al. in 2013 after a thorough molecular evaluation and some species formerly belonging to the genus *Clathrina* were placed under this newly erected genus. However, the genus name was already taken by a gastropod and in 2021, the new genus name *Ernsta* was proposed to replace *Ernstia* including *Ernstia naturalis*, thus presently known as *Ernsta naturalis* (Klautau et al., 2021) [12].

In our continuing search for bioactive metabolites from marine invertebrates, *Ernsta naturalis* (Van Soest and De Voogd, 2015, 2018) collected in Rodrigues (Indian Ocean) was investigated [13]. The organic crude extract of this animal exhibited moderate inhibitory activity against proteasome and tyrosinase. Our chemical investigation of this extract led to the isolation of a novel aminopyrimidone alkaloid compound, ernstine A (**1**), seven aminoimidazole alkaloid compounds, phorbatopsins D–E (**2**–**3**), calcaridine C (**4**), naamines H-I (**5**, **7**), naamidines J-K (**6**, **8**), together with the known thymidine (**9**). We report herein the purification and the structure elucidation by spectroscopic analysis including HRESIMS and 2D NMR for the new compounds (**1**–**8**) and comparison with published data for thymidine (**9**) [14]. In order to improve the investigation of this unstudied calcareous marine sponge, a metabolomic study by molecular networking (MN) was conducted. A molecular network is a computational strategy that may help visualization and interpretation of complex data from MS analysis, as crude extracts analysis, by organizing tandem mass spectrometry data through spectral similarities [15]. In MN, MS/MS data are represented in a graphical form, where an ion with an associated fragmentation spectrum is represented by a node and the links between two nodes indicate similarities between the two spectra. Consequently, only compounds with close fragmentation pathways will be linked together and will be grouped in clusters, highlighting families of compounds with the same skeletons. This representation can be particularly useful for the propagation of annotations from isolated compounds to other molecules of the crude extracts with close MS/MS data and so enhance the dereplication of the extract.

## 2. Results and Discussion

### 2.1. Characterization of New Compounds

The CH_2_Cl_2_-MeOH extract was first subjected to reverse-phase silica gel column chromatography to yield nine fractions. The fractions were subjected to SPE, repetitive reverse-phase semi-preparative, and analytical HPLC to yield nine compounds (**1**–**9**) (Figure 1). Eight were new: ernstine A (**1**), phorbatopsins D-E (**2**, **3**), calcaridine C (**4**), naamines H-I (**5**, **7**), naamidines J-K (**6**, **8**) described below and in addition, one other known compound was identified as thymidine (**9**) by comparison with published spectroscopic data.

Ernstine A (**1**) was obtained as a yellow solid. The molecular formula, C_19_H_19_N_3_O_3_, was established from HRESIMS molecular ion peak at *m/z* 338.1497 [M+H]^+^. Analysis of the 1D and 2D ^1^H, and ^13^C NMR data for **1** (CD_3_OD, Table 1) revealed resonances and correlations consistent with those of two para-substituted phenol, but not linked with an aminoimidazolone moiety, such as calcarine A, but linked with an aminopyrimidone moiety (Figure 2). The ^1^H NMR spectrum of **1** recorded in CD_3_OD showed the presence of two AA’BB’ spin systems at δH 6.94 and 6.74 (each 2H, d, J = 8.8 Hz) and at δH 7.08 and 6.88 (each 2H, d, J = 8.8 Hz), one singlet at δH 3.78 (3H, s), one singlet at δH 3.73 (3H, s), and one singlet at δH 3.56 (2H, s). Analysis and comparison of HSQC and HMBC correlations pointed out one methylene C-7 (δH 3.56; δC 39.6), two oxymethyl carbons C-12, C-17 (δH 3.73, 3.78; δC 55.4, 55.4), eight aromatic methines C-9, C-9′, C-10, C-10′, C-14, C-14′, C-15, C-15′, (δH 2 × 6.94, 2 × 6.74, 2 × 7.08, 2 × 6.88; δC 2 × 129.9, 2 × 114.1, 2 × 132.2, 2 × 114.1) of four chemically equivalent spin-pairs indicating a symmetry in the aromatic moieties, four quaternary aromatic carbons C-8, C-11, C-13, C-16 (δC 130.4, 158.4, 126.5, 159.1), two quaternary sp^2^ carbons due to the double bond C-5, C-6 (δC 115.9, 154.8), a guanidine-like carbon C-2 (δC 158.4) and one amide carbonyl group C-4 (δC 163.0). The COSY correlations between H-9 and H-10 in addition of the HMBC correlations between H-9 and C-7, C-9′and C-11 and between H-10 and C-8 and C-10′ indicated a symmetry and the presence of a para-phenolic group linked to the methylene C-7 in C-8. The HMBC correlations between H-12 and C-11 confirmed the substitution of the aromatic moiety in C-11 by the methoxy group C-12. In the same way, the COSY correlations between H-14 and H-15 in addition to the HMBC correlations between H-14 and C-14′ and C-16, between H-15 and C-13, and C-15′ and between H-17 and C-16 also revealed the presence of symmetry and a second para-phenolic moiety substituted by the methoxy group C-17 in C-16. The connection of the different moieties is confirmed by the NOE correlation cross-peaks (Figure 2). The HMBC correlation between H-14 and C-5 allowed linking the second nonprotonated carbon of this moiety C-13 to the quaternary sp^2^ carbon C-5. HMBC correlations between H-7 and C-2, C-4, C-5, and C-6, in addition to the molecular formula, C_19_H_19_N_3_O_3_ indicating 12 degrees of insaturations, revealed the presence of the aminopyrimidone moiety. This is the first report of an aminopyridine alkaloid from a calcareous sponge.

Phorbatopsin D (**2**) was obtained as a yellow solid. The molecular formula, C_11_H_13_N_3_O_3_, was established from HRESIMS molecular ion peak at *m/z* 236.1029 [M+H]^+^. Analysis of the 1D and 2D ^1^H, and ^13^C NMR data for **2** (CD_3_OD, Table 2) revealed resonances and correlations (Figure 2) consistent with those of a para-substituted phenol linked with an aminoimidazolone group, such as phorbatopsin B and C [16]. The ^1^H NMR spectrum of **2** recorded in CD_3_OD showed the presence of an AA’XX’ spin system at δH 7.01 and 6.64 (each 2H, d, J = 8.4 Hz), one singlet at δH 3.17 (3H, s), and one AB spin system at δH 3.00 and 2.95 (each 1H, d, J = 13.7 Hz). Analysis of the HSQC and HMBC correlations and the comparison with the latter compounds pointed out one methylene C-6 (δH 3.00 and 2.95; δC 41.9), one oxymethyl C-11 (δH 3.17; δC 51.6), four aromatic methines C-8, C-8′, C-9, C-9′ (δH 2 × 7.01, 2 × 6.64; δC 2 × 132.4, 2 × 115.9) of two chemically equivalent spin-pairs indicating a symmetry in the aromatic moiety, two quaternary aromatic carbons C-7, C-10 (δC 125.7, 157.1), one quaternary carbon of hemiaminal C-5 (δC 95.2) and one amide carbonyl group C-4 (δC 188.3). Compound **2** was different from phorbatopsin C by the presence of the oxymethyl C-11 and the quaternary carbon of hemiaminal C-5 instead of one aminomethine. The COSY correlations between H-8 and H-9 in addition to the HMBC correlations between H-8 and C-6, C-8′ and C-10, and between H-9 and C-7 and C-9′ indicated a symmetry and the presence of a para-phenolic group linked to the methylene C-6 in C-7. The HMBC correlation between H-6 and C-4, C-5, C-7, and C-8 indicated the substitution of the methylene by the para-phenolic core and by the quaternary carbon of hemiaminal C-5. The HMBC correlation between H-11 and C-5 indicated the substitution of the quaternary carbon of hemiaminal C-5 by a methoxy group. The chemical shift of the amide carbonyl group C-4 (δC 188.0) of phorbatopsin D (**2**) was close to that of phorbatopsin C (δC 188.7) but 17.0 ppm higher than that of phorbatopsin B (δC 171.0) described by Nguyen et al. [16]; this difference could be explained by the annular tautomerism (as classified by Katritzky and Lagowski [17]) of the aminoimidazolone moiety. The chemical shift of C-4 in phorbatopsin D and phorbatopsin C corresponded to the C-4 of the tautomer **a** (Figure 3), whereas the chemical shift of C-4 in phorbatopsin B corresponds to the C-4 of the tautomer **b** [18,19]. Indeed, Krawczyk et al. [18], had demonstrated that for creatinines substituted at position 5 with an electron-withdrawing substituent, the amine tautomer **a** is preferred in a polar solvent. Compound **2** was named phorbatopsin D according to phorbatopsin B and C reported in 2012 [16].

Phorbatopsin E (**3**) was obtained as a yellow solid. The molecular formula, C_12_H_15_N_3_O_3_, was established from HRESIMS molecular ion peak at *m/z* 250.1188 [M+H]^+^. Analysis of the 1D and 2D ^1^H, and ^13^C NMR data for **3** (CD_3_OD, Table 2) revealed resonances and correlations consistent with those of a para-substituted phenol linked with an aminoimidazolone group, such as phorbatopsin D (**2**). Compound **3** was different from **2** by the presence of the oxymethyl C-12 (δH 3.73; δC 55.2) instead of an alcohol group. This is confirmed by the HMBC correlations between H-12 and C-10 and NOE correlations between H-12 and H-9/H-9′.

Calcaridine C (**4**) was obtained as a yellow solid. The molecular formula, C_18_H_19_N_3_O_4_, was established from HRESIMS molecular ion peak at *m/z* 342.1449 [M+H]^+^. Analysis of the 1D and 2D ^1^H, and ^13^C NMR data for **4** (CD_3_OD, Table 3) revealed resonances and correlations consistent with those of two para-substituted phenol linked with an aminoimidazolone moiety, such as calcarine A [3]. Compound **4** was different from calcaridine A by the presence of an alcohol group in C-15 instead of a methoxy group and the substitution of *N*-1 which was substituted by a proton instead of a methyl. Moreover, the chemical shift of the amide carbonyl group C-4 (δC 189.8) of calcaridine C (**4**) was 15.8 ppm higher than that of calcaridine A (δC 174.0) described by Edrada et al. [3], calcaridine C corresponded to the tautomer **a** (Figure 3) whereas calcaridine A corresponded to tautomer **b**. This difference in isomeric protonation states could be explained by differences in the isolation protocol, herein all the compounds had been isolated in acidic conditions (0.1% formic acid) while Edradra et al. had isolated calcaridine A without acid. Compound **4** was named calcaridine C according to calcaridine A reported in 2003 [3] and calcaridine B reported in 2018 [20].

Naamine H (**5**) was obtained as a yellow solid. The molecular formula, C_18_H_19_N_3_O_2_, was established from HRESIMS molecular ion peak at *m/z* 310.1544 [M+H]^+^. Analysis of the 1D and 2D ^1^H, and ^13^C NMR data for **5** (CD_3_OD, Table 4) showed that it was closely related to naamine A to G [4,21,22,23,24,25,26,27], namely resonances and correlations consistent with those of two para-substituted phenol linked with a 2-aminoimidazole group (Figure 2). Compound **5** was different from naamine A by the lack of a methyl group attached to the N-3 of the 2-aminoimidazole ring.

Naamidine J (**6**) was obtained as a yellow solid. The molecular formula, C_22_H_21_N_5_O_4_, was established from HRESIMS molecular ion peak at *m/z* 420.1664 [M+H]^+^. Analysis of the 1D and 2D ^1^H, and ^13^C NMR data for **6** (CD_3_OD, Table 5) showed that it was closely related to naamine H (**5**) and to naamidines A to I [22,23,28,29]. Namely, as naamine H (**5**), resonances and correlations were consistent with those of two para-substituted phenol linked with a 2-aminoimidazole ring but herein this 2-aminoimidazole ring was also linked to a hydantoin ring. The substitution of the benzyl rings was the same as naamine H (**5**), namely, one alcohol function and one methoxy group and the substitution of the hydantoin ring was the same as naamidine A, by one methoxy group on nitrogen.

Naamine I (**7**) and Naamidine K (**8**) were obtained as a yellow solid mixture. The molecular formula, C_17_H_17_N_3_O_2_, of naamine I was established from HRESIMS molecular ion peak at *m/z* 296.1389 [M+H]^+,^ and the molecular formula, C_21_H_19_N_5_O_4_, of naamidine K was established from HRESIMS molecular ion peak at *m/z* 406.1502 [M+H]^+^. Analysis of the 1D and 2D ^1^H, and ^13^C NMR data of the mixture of **7** and **8** (CD_3_OD, Table 4 and Table 5) showed that it was closely related to naamine H (**5**) and to naamidine J (**6**). They were only differing by the substitution of the benzyl rings by two hydroxyls instead of one hydroxyl and one methoxy group.

### 2.2. Dereplication of the Crude Extract

To obtain the first molecular fingerprint of the unstudied Rodrigues calcareous marine sponge *Ernsta naturalis*, the CH_2_Cl_2_-MeOH extract was profiled by HPLC-HRMS/MS. These data were subsequently processed by GNPS [30]. Beforehand, the mass spectra of the eight new molecules isolated and characterized by NMR were submitted to the library of the GNPS (Accession codes of the isolated compounds). These molecules could thus be reported directly in the molecular network of the extract. The molecular network (Figure 4) contains 167 nodes including 111 clustered molecules. The isolated molecules are distributed in two clusters of interest. Naamine and naamidine derivatives grouped together with ernstine A (**1**) in the first cluster 1 of twenty-three molecules (Figure 5). Phorbatopsin derivatives (**2**, **3**) and calcaridine C (**4**) grouped together in cluster 2 of twenty-one molecules (Figure 6).

The naamidine derivatives cluster was characterized by the presence of neutral loss (C_7_H_8_O and C_6_H_6_O) which correspond to the methoxylated or hydroxylated phenolic group. In addition to the loss of neutral, characteristic ions were observed, as 160.0758 *m*/*z* [C_10_H_10_NO]^+^ which determines the presence of the phenolic group with the 2-aminoimidazole moiety. Moreover, the presence of the loss of neutral (C_3_H_3_NO_2_ or C_3_H_4_N_2_O or C_4_H_6_N_2_O) corresponding to the fragmentation in the 3-methyl-imidazolidin-4-one ring helps to indicate whether the group corresponds to either 3-methylimidazolidine-2,4-dione (as naamidine), or 2-imino-3-methyl-imidazolidin-4-one or 2-methylimino-3-methyl-imidazolidin-2-one. With these elements, it is possible to propagate the annotations of cluster 1 by characterizing other nodes. Thus, structural hypotheses of nine additional molecules have been proposed (Table 6), in addition to the spectral confirmation of the five compounds already isolated and characterized by NMR. With this methodology, two nodes seem to correspond to naamine D isomers [23] and one node to naamidine D [22], two molecules isolated from the calcareous sponge Leucetta, eight other nodes seem to correspond to new molecules and for the last seven ones, the hypothesis was too uncertain to propose a structural hypothesis.

Using the same approach, the cluster of phorbatopsin derivatives could be partially characterized. This cluster is divided into two parts. The first one contains two nodes identified as calcaridine C (**4**) with similar spectra but different retention times. Calcaridine C is characterized by the ions 107.0490 *m/z* [C_7_H_7_O]^+^ and 137.0597 *m/z* [C_8_H_9_O_2_]^+^, characterized by the phenolic group. However, the fragmentation is distinguished by a loss of neutrality in C_8_H_10_O_2_. This part of the cluster possesses many nodes with the same masses and similar spectra probably due to the presence of isomers. It complicates the interpretation of compound spectra, so no additional annotation was added in this part of the cluster. Finally, phorbatopsin D and E (**2**, **3**) were projected in the second part of the cluster where six other molecules could be proposed (Table 7). For these molecules, the losses of neutrals CO, CH_4_O, and C_2_H_2_N_2_O due to fragmentation in the 2-aminoimidazolin-4-one cycle and the characteristic ions 107.0492 *m/z* [C_7_H_7_O]^+^ and 121.0646 *m/z* [C_8_H_9_O]^+^ which correspond to the phenolic group without and with a methoxy are found. Two known molecules with the same 2-aminoimidazolin-4-one moiety were proposed to be phorbatopsins A and C, along with three new compounds. Only one molecule with a different moiety has been proposed, leucettamine C, with a loss of C_3_H_6_N_2_O corresponding to a 2-imino-3-methyl-imidazolidin-4-one moiety.

### 2.3. Biosynthetic Pathway

Even if a clear definition of biosynthetic origin of the 2-aminoimidazole alkaloids from the calcareous sponges has not been established at the present time, different hypotheses of biological pathway have been proposed but no experimental confirmation has been reported [31]. Crews had proposed a biosynthesis pathway including an intermediate with one phenyl ring coming from guanidine and *p*-hydroxyphenylpyruvic acid [31,32]. The presence in the crude extract of *Ernsta naturalis* of a compound with an ion peak in HRESIMS at *m/z* 220.1078 [M+H]^+^ corresponding to the molecular formula of the intermediate *p*-methoxyphorbatopsin C, in addition to the presence of both aminoimidazole alkaloids with one phenyl ring and aminoimidazole alkaloids with two phenyl rings are in agreement with this proposal.

## 3. Materials and Methods

### 3.1. General Experiment Procedures

Optical rotations were measured on a MCP 200 Anton Paar modular circular polarimeter at 25 °C (MeOH, *c* in g/100 mL). ^1^H and ^13^C NMR data were acquired with a Bruker Avance II+—600 MHz spectrometer equipped with a TCI Cryoprobe at 300 K with 2 mm o.d. Match NMR tubes. Chemical shifts were referenced using the corresponding solvent signals (δ_H_ 3.31 and δ_C_ 49.00 for CD_3_OD). The spectra were processed using 1D and 2D NMR MNova software (Version No. 14.1.1-24571, Mestrelab Research S. L., Santiago de Compostela, Spain). HRESIMS spectra were recorded using a Waters SYNAPT G2 HDMS mass spectrometer (Waters, Guyancourt, France).

The sponge was lyophilized with Cosmos −80 °C CRYOTEC. MPLC separations were carried out on a Buchi Sepacore flash system C-605/C-615/C-660 and glass column (230 × 15 mm i.d.) packed with Acros Organics C18-RP, 23%C, silica gel (40−63 μm). Precoated TLC sheets of silica gel 60, Alugram SIL G/UV254 were used, and spots were visualized on the basis of the UV absorbance at 254 nm and by heating silica gel plates sprayed with formaldehyde−sulfuric acid or Dragendorff reagents. HPLC analyses were carried out using a Phenomenex Gemini C_18_ (150 × 4.6 mm i.d., 3 μm) column and were performed on a Thermo Scientific Dionex Ultimate 3000 system equipped with a photodiode array detector and a Corona detector with Chromeleon software. Semi-preparative HPLC was carried out using a Phenomenex Geminin C_18_ (250 × 10 mm i.d., 5 μm) column and was performed on a Thermo Scientific Dionex Ultimate 3000 system equipped with a photodiode array detector. All solvents were analytical or HPLC grade and were used without further purification.

### 3.2. Animal Material

The sponge *Ernsta naturalis* (phylum Porifera, class Calcarea, order Clathrinida, family Clathrinidae) was collected in October 2016 in Passe Balidirou, Rodrigues (19°40.098′ S, 63°27.784′ E at 12–15 m depth). One voucher specimen (RMNH Por. 11633) was deposited in the sponge collection of Naturalis Biodiversity Center, the Netherlands. Sponge samples were frozen immediately and kept at −20 °C until processed.

### 3.3. Extraction and Isolation

The frozen sponge (53.5 g, dry weight) was chopped into small pieces, lyophilized and extracted exhaustively by maceration with CH_2_Cl_2_-MeOH (1:1 *v:v*) (2 × 1.5 L, each 24 h) at room temperature. After evaporating the solvents under reduced pressure, a brown, oily residue (3.07 g) was obtained. The extract was then subjected to MPLC over C18-RP silica gel in a glass column (230 × 15 mm i.d.), eluting with a combination of water and MeOH of decreasing polarity (15 mL min^−1^). Nine fractions were obtained: F0 eluted with H_2_O-MeOH (95:5) over 5 min; F1 eluted with H_2_O−MeOH (95:5) over 5 min; F2 eluted with H_2_O-MeOH (75:25) over 5 min, F3 eluted with H_2_O−MeOH (50:50) over 5 min, F4 eluted with H_2_O-MeOH (25:75) over 5 min; F5 to F8 eluted with H_2_O-MeOH (95:5) over 20 min.

Fraction F0 (1.21 g) was fractionated by C-18 SPE, eluted with a combination of water and MeOH of decreasing polarity and three subfractions were obtained (SF0–SF2).

Subfraction SF1 (57 mg) was subjected to semipreparative HPLC (Phenomenex Geminin C18 column, 250 × 10 mm i.d., 5 µm., 4.5 mL min^−1^ gradient elution with 5% ACN-H_2_O (+0.1% formic acid) over 5 min, then 5% to 15% ACN-H_2_O (+0.1% formic acid) over 30 min and 15% ACN-H_2_O (+0.1% formic acid) over 10 min; UV 220, 280 nm) to provide pure compounds 2 (phorbatopsin D, 1.6 mg), 3 (phorbatopsin E, 2.7 mg), 4 (calcaridine C, 2.1 mg) and 9 (thymidine, 4.2 mg).

Fraction F2 (64 mg) was subjected to semipreparative HPLC (Phenomenex Geminin C18 column, 250 × 10 mm i.d., 5 µm., 4.5 mL min^−1^ gradient elution with 12% ACN-H_2_O (+0.1% formic acid) over 5 min, then 12% to 35% ACN-H2O (+0.1% formic acid) over 35 min and 35% ACN-H_2_O (+0.1% formic acid) over 5 min; UV 220, 280 nm) to obtain 11 subfractions (F2SF1-F2SF11). Pure compounds 5 (naamine H, 0.9 mg), 7 (naamidine J, 1.6 mg), and 1 (ernstine A, 1.1 mg) were obtained in the subfractions, F2SF7, F2SF11 and F2SF9, respectively.

Subfraction F2SF10 (3.1 mg) was subjected to semipreparative HPLC (Phenomenex Geminin C18 column, 250 × 10 mm i.d., 5 µm., 4.5 mL min^−1^ isocratic elution with 26% ACN-H_2_O (+0.1% formic acid) over 20 min; UV 220, 280 nm) to provide one mixture of compound 6 and compound 8 (naamine I and naamidine K, 1.4 mg) and a pure compound 1 (ernstine A, 1.0 mg).

*Ernstine A* (**1**). Yellow oil, [α]_D_^25^ 0.0 (*c 0.1*, MeOH); ^1^H and ^13^C NMR, see Table 1; HRESIMS *m/z* 338.1497 [M + H]^+^ (calcd for C_19_H_20_N_3_O_3_^+^, 338.1499).

*Phorbatopsin D* (**2**). Yellow oil, ^1^H and ^13^C NMR, see Table 2; HRESIMS *m/z* 236.1029 [M + H]^+^ (calcd for C_11_H_14_N_3_O_3_^+^, 236.1030).

*Phorbatopsin E* (**3**). Yellow oil, [α]_D_^25^ 0.0 (*c 0.1*, MeOH); ^1^H and ^13^C NMR, see Table 2; HRESIMS *m/z* 250.1188 [M + H]^+^ (calcd for C_12_H_16_N_3_O_3_^+^, 250.1186).

*Calcaridine C* (**4**). Yellow oil, [α]_D_^25^ 0.0 (*c 0.1*, MeOH); ^1^H and ^13^C NMR, see Table 3; HRESIMS *m/z* 342.1449 [M + H]^+^ (calcd for C_18_H_20_N_3_O_4_^+^, 342.1448).

*Naamine H* (**5**). Yellow oil, ^1^H and ^13^C NMR, see Table 4; HRESIMS *m/z* 310.1544 [M + H]^+^ (calcd for C_18_H_20_N_3_O_2_^+^, 310.1550).

*Naamidine J* (**6**). Yellow oil, [α]_D_^25^ 0.0 (*c 0.1*, MeOH); ^1^H and ^13^C NMR, see Table 5; HRESIMS *m/z* 420.1664 [M + H]^+^ (calcd for C_22_H_22_N_5_O_4_^+^, 420.1666).

*Naamine I* (**7**). Yellow oil, ^1^H and ^13^C NMR, see Table 4; HRESIMS *m/z* 296.1389 [M + H]^+^ (calcd for C_17_H_18_N_3_O_2_^+^, 296.1394).

*Naamidine K* (**8**). Yellow oil, [α]_D_^25^ 0.0 (*c 0.1*, MeOH); ^1^H and ^13^C NMR, see Table 5; HRESIMS *m/z* 406.1502 [M + H]^+^ (calcd for C_21_H_20_N_5_O_4_^+^, 406.1510).

### 3.4. UHPLC/HRMS/MS

Crude extract was analyzed on an Ultimate 3000 UHPLC system (Dionex, Germering, Germany) hyphenated with Impact II high resolution quadrupole time-of-flight (QqTOF) equipped with an electrospray ionization (ESI) source (Bruker Daltonics, Bremen, Germany). Separation of extract was achieved on a Luna C18 column (150 mm × 2.1 mm, 1.6 µm) with an injection volume of 2 µL. A binary solvent system was used as mobile phase, solvent A consisting of water with 0.1% (*v*/*v*) formic acid and solvent B consisting of acetonitrile with 0.1% (*v*/*v*) formic acid. The flow rate was 0.6 mL min^−1^, and a gradient was applied: from 10% to 100% of B in 15min. The acquisition was carried out in ESI positive ionization mode with a range of 50–1200 Da. The capillary voltage was maintained at 3 kV, the gas flow to the nebulizer was set at 3.5 bars, the drying temperature was 200 °C, and the drying gas flow was 4 L min^−1^. The collision-induced dissociated (CID) energy was applied from 20 to 40 eV.

Data were analyzed using Bruker Data Analysis 4.4 software. The data were processed using MetaboScape 4.0 (Bruker Daltonics, Bremen, Germany). A mgf file was submitted to the GNPS (Global Natural Product Social Networking) web-based platform for generating MS based molecular network [33]. The following parameters were applied to create the molecular network. The mass tolerance was 0.01 Da for precursor and fragment ions. Minimum score was 0.6 between two MS/MS spectra to be connected. The minimum number of common fragment ions between two MS/MS spectra was 3. The nearly identical MS/MS spectra were not merged into consensus MS/MS spectrum. A node was allowed to connect to a maximum of 10 nodes. A cluster can have a maximum of 100 nodes. The spectra in the network were then compared with GNPS spectral libraries [31]. Each MS^2^ spectrum of the seven isolated compounds was assigned an individual accession number on the GNPS (Appendix A). The molecular networking was visualized using Cytoscape (ver. 3.6.0). The obtained molecular network can be accessed at: https://gnps.ucsd.edu/ProteoSAFe/status.jsp?task=527472b15d4247dfad0534aa80f7ebfa, accessed on 8 October 2022.

## 4. Conclusions

In conclusion, one novel aminopyrimidone alkaloid compound, ernstine A (**1**), seven new aminoimidazole alkaloid compounds, phorbatopsins D-E (**2**, **3**), calcaridine C (**4**), naamines H–I (**5**, **7**), naamidines J–K (**6**, **8**) were isolated from a CH_2_Cl_2_-MeOH extract from *Ernsta naturalis* along with the known thymidine (**9**). To improve the investigation of this unstudied calcareous marine sponge, a metabolomic study by molecular networking was conducted. This strategy, based on the interpretation of MS/MS spectra of other compounds grouped in the same clusters than the isolated ones due to their structural feature similarities, allowed us to propose structural hypotheses for 16 compounds, 5 known and 11 potentially new.

## Figures and Tables

**Figure 1 marinedrugs-20-00637-f001:**
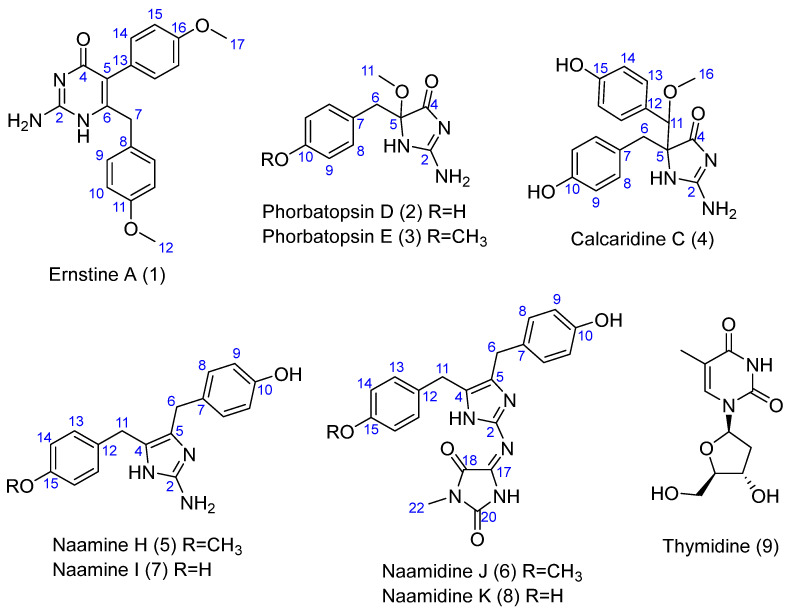
Chemical structures of compounds **1**–**9**.

**Figure 2 marinedrugs-20-00637-f002:**
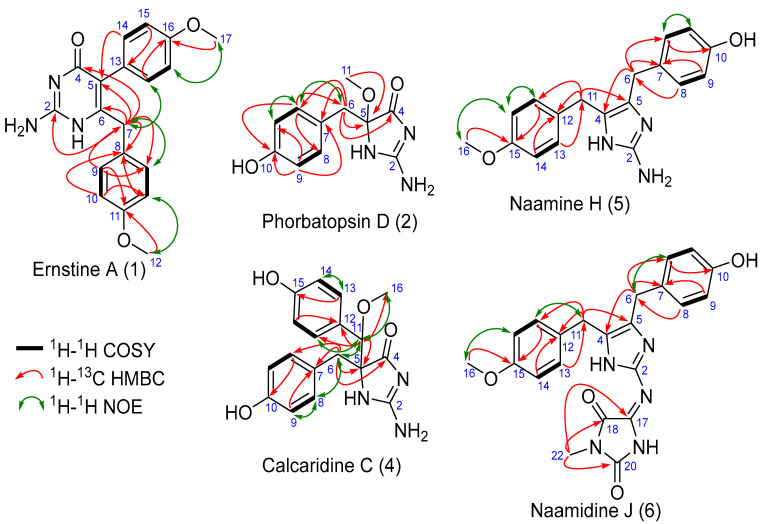
Key COSY, HMBC and NOE correlations for compounds **1**, **2**, **4**, **5** and **6**.

**Figure 3 marinedrugs-20-00637-f003:**
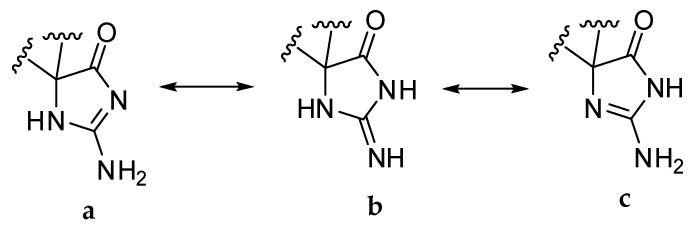
Three tautomeric forms (**a**–**c**) of the 5-substituted aminoimidazolone moiety.

**Figure 4 marinedrugs-20-00637-f004:**
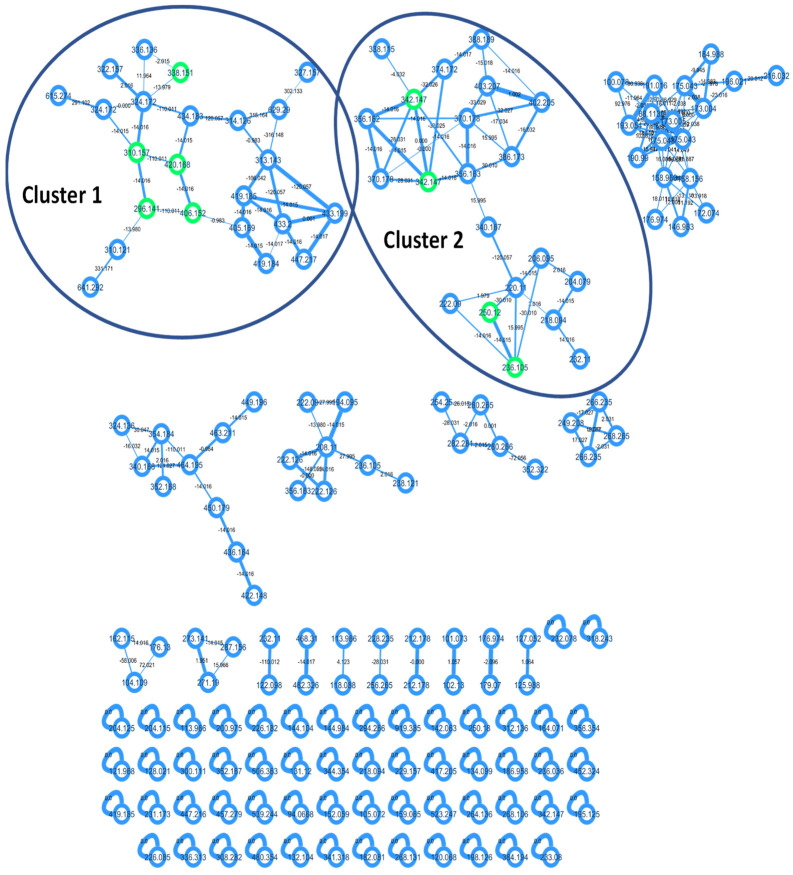
Molecular network of *Ernsta naturalis* crude extract. Isolated molecules are in green in the molecular network.

**Figure 5 marinedrugs-20-00637-f005:**
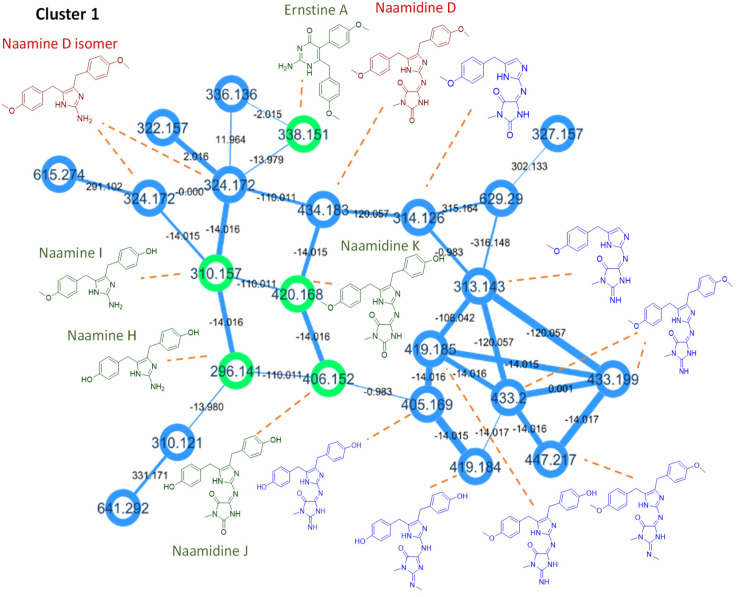
Cluster 1 of the molecular network of *Ernsta naturalis* crude extract. Isolated molecules are in green in the molecular network, the proposals of known molecules in red, and the proposals of new molecules in blue.

**Figure 6 marinedrugs-20-00637-f006:**
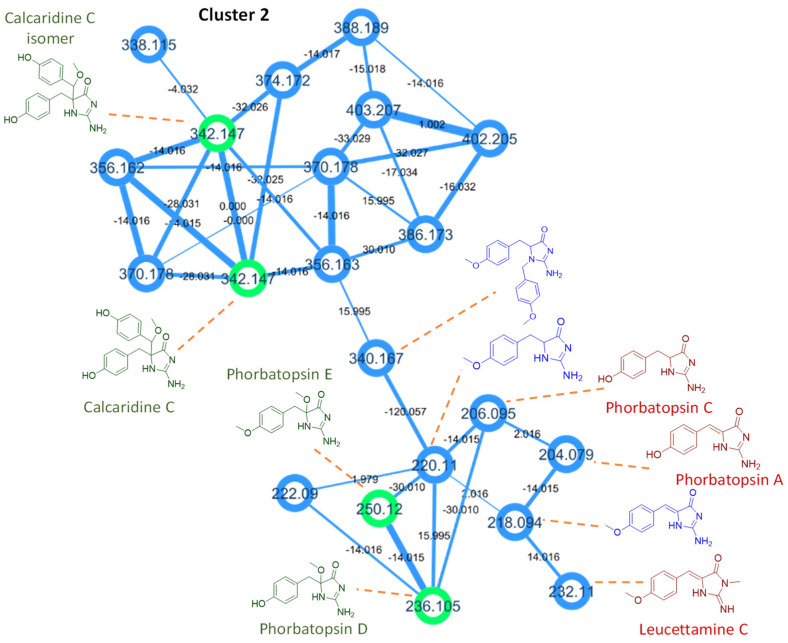
Cluster 2 of the molecular network of *Ernstia naturalis* crude extract. Isolated molecules are in green in the molecular network, the proposals of known molecules in red, and the proposals of new molecules in blue.

**Table 1 marinedrugs-20-00637-t001:** The 1D and 2D NMR spectroscopic data (^1^H, ^13^C 600/150 MHz, CD_3_OD) for ernstine A (**1**).

n°	δC, Type	δH (*J* in Hz)	COSY (^1^H-^1^H)	HMBC (^1^H-^13^C)	NOESY (^1^H-^1^H)
2	158.4, C	-	-	-	-
4	163.0, C	-	-	-	-
5	115.9, C	-	-	-	-
6	154.8, C	-	-	-	-
7	39.6, CH_2_	3.56, 2H, s	-	2, 4, 5, 6, 8, 9, 9′	9, 9′, 14, 14′
8	130.4, C	-	-	-	-
9, 9′ *	129.9, CH	6.94, 2H, d (8.8)	10, 10′	7, 9, 9′, 11	7
10, 10′ *	114.1, CH	6.74, 2H, d (8.8)	9, 9′	8, 10, 10′	12
11	158.4, C	-	-	-	-
12	55.4, CH_3_	3.73, 3H, s	-	11	10, 10′
13	126.5, C	-	-	-	-
14, 14′ *	132.2, CH	7.08, 2H, d (8.8)	15, 15′	5, 14, 14′, 16	7
15, 15′ *	114.2, CH	6.88, 2H, d (8.8)	14, 14′	13, 15, 15′	17
16	159.1, C	-	-	-	-
17	55.4, CH_3_	3.78, 3H, s	-	16	15, 15′

*** Chemically equivalent spin-pairs.

**Table 2 marinedrugs-20-00637-t002:** The 1D and 2D NMR spectroscopic data (^1^H, ^13^C 600/150 MHz, CD_3_OD) for phorbatopsin D (**2**) and E (**3**).

	Phorbatopsin D (2)			Phorbatopsin E (3)		
n°	δC, Type	δH (*J* in Hz)	HMBC (^1^H-^13^C)	δC, Type	δH (*J* in Hz)	HMBC (^1^H-^13^C)
2	-	-	-	-	-	-
4	188.3, C	-	-	186.3, C	-	-
5	95.2, C	-	-	95.0, C	-	-
6	41.9, CH_2_	2.95, 1H, d (13.7) 3.00, 1H, d (13.7)	4, 5, 7, 8, 8′	41.4, CH_2_	3.01, 1H, d (13.6) 3.05, 1H, d (13.6)	4, 5, 7, 8, 8′
7	125.7, C	-	-	126.8, C	-	-
8, 8′ *	132.4, CH	7.01, 2H, d (8.4)	6, 8, 8′, 10	132.1, CH	7.11, 2H, d (8.5)	6, 8, 8′, 10
9, 9′ *	115.9, CH	6.64, 2H, d (8.4)	7, 9, 9′	114.3, CH	6.78, 2H, d (8.6)	7, 9, 9′, 10
10	157.1, C	-	-	160.1, C	-	-
11	51.6, CH_3_	3.17, 3H, s	5	51.1, CH_3_	3.19, 3H, s	5
12	-	-	-	55.2, CH_3_	3.73, 3H, s	10

* chemically equivalent spin-pairs.

**Table 3 marinedrugs-20-00637-t003:** The 1D and 2D NMR spectroscopic data (^1^H, ^13^C 600/150 MHz, CD_3_OD) for calcaridine C (**4**).

n°	δC, Type	δH (*J* in Hz)	COSY (^1^H-^1^H)	HMBC (^1^H-^13^C)
2	-	-	-	-
4	189.8, C	-	-	-
5	75.3, C	-	-	-
6	39.5, CH_2_	2.35, 1H, d (13.8) 2.94, 1H, d (13.8)	6	4, 5, 7, 8, 8′
7	126.4, C	-	-	-
8, 8′ *	132.2, CH	6.86, 2H, d (8.2)	9, 9′	6, 8, 8′, 10
9, 9′ *	115.9, CH	6.58, 2H, d (8.2)	8, 8′	7, 9, 9′, 10
10	157.4, C	-	-	-
11	86.2, CH	4.35, 1H, s	-	5, 12, 13, 13′, 16
12	128.3, C	-	-	-
13, 13′ *	130.6, CH	7.25, 2H, d (8.3)	14, 14′	11, 13, 13′, 15
14, 14′ *	116.4, CH	6.86, 2H, d (8.3)	13, 13′	12, 14, 14′, 15
15	159.0, C	-	-	-
16	57.1, CH_3_	3.15, 3H, s	-	11

* Chemically equivalent spin-pairs.

**Table 4 marinedrugs-20-00637-t004:** The 1D and 2D NMR spectroscopic data (^1^H, ^13^C 600/150 MHz, CD_3_OD) for Naamines H (**5**) and I (**7**).

	Naamine H (5)			Naamine I (7)		
n°	δC, Type	δH (*J* in Hz)	HMBC (^1^H-^13^C)	δC, Type	δH (*J* in Hz)	HMBC (^1^H-^13^C)
2	-	-	-	-	-	-
4	123.8, C	-	-	126.4, C	-	-
5	123.8, C	-	-	126.4, C	-	-
6	29.3, CH_2_	3.73, 2H, s	4, 5, 7, 8, 8′	29.7, CH_2_	3.80, 2H, s	4, 5, 7, 8, 8′
7	129.9, C	-	-	128.4, C	-	-
8, 8′ *	129.8, CH	6.98, 2H, d (8.5)	6, 8, 8′, 10	129.3, CH	6.95, 2H, d (8.3)	6, 8, 8′, 9, 9′, 10
9, 9′ *	115.8, CH	6.70, 2H, d (8.5)	7, 9, 9′	115.6, CH	6.71, 2H, d (8.5)	7, 9, 9′
10	156.9, C	-	-	156.1, C	-	-
11	29.3, CH_2_	3.76, 2H, s	4, 5, 12, 13, 13′	29.7, CH_2_	3.80, 2H, s	4, 5, 12, 13, 13′
12	131.3, C	-	-	128.4, C	-	-
13, 13′ *	129.8, CH	7.08, 2H, d (8.6)	11, 13, 13′, 15	129.3, CH	6.95, 2H, d (8.3)	11, 13, 13′, 15
14, 14′ *	114.4, CH	6.84, 2H, d (8.7)	12, 14, 14′	115.6, CH	6.71, 2H, d (8.5)	12, 14, 14′
15	159.6, C	-	-	156.1, C	-	-
16	54.9, CH_3_	3.76, 3H, s	-	-	-	-

* chemically equivalent spin-pairs.

**Table 5 marinedrugs-20-00637-t005:** The 1D and 2D NMR spectroscopic data (^1^H, ^13^C 600/150 MHz, CD_3_OD) for naamidines J (**6**) and K (**7**).

	Naamidine J (6)			Naamidine K (8)		
n°	δC, Type	δH (*J* in Hz)	HMBC (^1^H-^13^C)	δC, Type	δH (*J* in Hz)	HMBC (^1^H-^13^C)
2	-	-	-	-	-	-
4	129.0, C	-	-	126.4, C	-	-
5	129.1, C	-	-	126.4, C	-	-
6	30.6, CH_2_	3.88, 2H, s	4, 5, 7, 8, 8′	29.7, CH_2_	3.80, 2H, s	4, 5, 7, 8, 8′
7	130.4, C	-	-	128.4, C	-	-
8, 8′ *	130.4 CH	6.97, 2H, d (8.2)	6, 8, 8′ 10	129.3, CH	6.93, 2H, d (8.5)	6, 8, 8′, 10
9, 9′ *	116.4, CH	6.69, 2H, d (8.2)	7, 9, 9′	115.6, CH	6.71, 2H, d (8.5)	7, 9, 9′
10	157.2, C	-	-	156.1, C	-	-
11	30.6, CH_2_	3.85, 3H, s	5, 13, 13′	29.7, CH_2_	3.80, 3H, s	4, 5, 13, 13′
12	131.7, C	-	-	128.4, C	-	-
13, 13′ *	130.4, CH	7.06, 2H, d (83)	11, 13, 13′, 15	129.3, CH	6.93, 2H, d (8.5)	11, 13, 13′ 15
14, 14′ *	115.1, CH	6.82, 2H, d (8.3)	14, 14′, 12	115.6, CH	6.71, 2H, d (8.5)	12, 14, 14′, 15
15	160.0, C	-	-	156.1, C	-	-
17	-	-	-	-	-	-
18	166.4, C	-	-	157.6, C	-	-
20	163.9, C	-	-	159.5, C	-	-
22	24.7, CH_3_	3.04, 3H, s	18, 20	24.4, CH_3_	3.06, 3H, s	18, 20
23	55.71, CH_3_	3.75, 3H, s	15			

* Chemically equivalent spin-pairs.

**Table 6 marinedrugs-20-00637-t006:** Tentative identification of cluster A compounds from extract of the Rodrigues calcareous marine sponge *Ernsta naturalis* by LC-ESI-MS/MS in the positive ion mode. The presence of characteristic ion on the MS² spectra was indicated by a X in the table.

RT (min)	Neutral Loss MS^2^					Ion Characteristic MS^2^	*m/z* [M+H]+	Raw Formula	Error (ppm)	Molecule Tentative Identification (INCHI Key)	Confidence Level
	C_6_H_6_O	C_7_H_8_O	C_3_H_3_NO_2_	C_3_H_4_N_2_O	C_4_H_6_N_2_O	C_10_H_10_N					
6.16	202.0975					X	296.1393	C_17_H_18_N_3_O_2_	0.3	Naamine H (KFOAYDULNDFMPQ-UHFFFAOYSA-N)	1
6.18	227.0925			321.1342		X	405.1664	C_21_H_21_N_6_O_3_	1.4	(BLKZRPHRHGCARC-UHFFFAOYSA-N)	3
6.2		205.0834		229.1081			313.1404	C_15_H_17_N_6_O_2_	1.2	(JIKZHJXIVNJXCB-UHFFFAOYSA-N)	3
6.52	227.0922				321.1337	X	419.1825	C_22_H_23_N_6_O_3_	0.3	(KSJXZPZLMWQEOL-UHFFFAOYSA-N)	3
6.92	216.1132	202.0974				X	310.1549	C_18_H_20_N_3_O_2_	0.3	Naamine I (QFSIYRFDLATYCH-UHFFFAOYSA-N)	1
6.94	241.1085	311.1247		335.1501		X	419.1825	C_22_H_23_N_6_O_3_	0.3	(AUUWFNXXXPZWIM-UHFFFAOYSA-N)	3
7.02		216.1128				X	324.1703	C_19_H_22_N_3_O_2_	1	Naamine D isomer (JIAXZLQLTAUEFZ-UHFFFAOYSA-N)	2
7.11	312.1089					X	406.1503	C_21_H_20_N_5_O_4_	1.6	Naamidine J (ZITLIVILDBHVPT-UHFFFAOYSA-N)	1
7.14		241.1084		349.1655			433.1981	C_23_H_25_N_6_O_3_	0.3	(CKFJLVUSNNSFGX-UHFFFAOYSA-N)	3
7.3		230.0921				X	338.1492	C_19_H_20_N_3_O_3_	2.2	Ernstine A (DDPTZQPAIVSDEH-UHFFFAOYSA-N)	1
7.39		206.0671	121.0508				314.1241	C_15_H_16_N_5_O_3_	2.1	(FJLZROPMRMSUSB-UHFFFAOYSA-N)	3
7.58		241.1078		349.1653			433.1972	C_23_H_25_N_6_O_3_	2.5	(CKFJLVUSNNSFGX-UHFFFAOYSA-N)	3
7.6		216.1128					324.1703	C_19_H_22_N_3_O_2_	0.3	Naamine D isomer (JIAXZLQLTAUEFZ-UHFFFAOYSA-N)	4
7.87		339.1564			349.1655		447.2138	C_24_H_27_N_6_O_3_	0.2	(MZCUSFHZTJHTHB-UHFFFAOYSA-N)	3
8.06	326.1246	312.1088	227.0926			X	420.1659	C_22_H_22_N_5_O_4_	1.7	Naamidine K (BIKAACVDYVDVHU-UHFFFAOYSA-N)	1
9.1		326.1241	241.1080				434.1819	C_23_H_24_N_5_O_4_	0.8	Naamidine D (CXGRXOLKKUWCFJ-UHFFFAOYSA-N)	2

**Table 7 marinedrugs-20-00637-t007:** Tentative identification of cluster B compounds from extract of the Rodrigues calcareous marine sponge *Ernsta naturalis* by LC-ESI-MS/MS in the positive ion mode. The presence of characteristic ion on the MS² spectra was indicated by a X in the table.

RT (min)	Neutral Loss MS^2^				Ion Characteristic MS^2^			*m/z* Measured [M+H]^+^	Raw Formula	Error (ppm)	Molecule Tentative Identification (INCHI Key)	Confidence Level
	CH_4_O	CH_3_NO	C_2_H_2_N_2_O	C_3_H_6_N_2_O	C_8_H_9_O_2_	C_8_H_9_O	C_7_H_7_O					
1.25			136.0755				X	206.0929	C_10_H_12_N_3_O_2_	−2.5	Phorbatopsin C (MFHHWOMFRHLQSF-UHFFFAOYSA-N)	2
1.86	204.0767		166.0862			X	X	236.1031	C_11_H_14_N_3_O_3_	−0.6	Phorbatopsin D (IQLRXEDGGLMGEF-UHFFFAOYSA-N)	1
2.14		159.0553	132.0444				X	204.0766	C_10_H_10_N_3_O_2_	0.9	Phorbatopsin A (PZMLZQIKCWTTJV-YVMONPNESA-N)	2
4.05			150.0914			X		220.1078	C_11_H_14_N_3_O_2_	1	Methoxy phorbatopsin C (JQRQEDSHZOMVAE-UHFFFAOYSA-N)	2
5.6	218.0920		180.1014			X		250.1181	C12H_16_N_3_O_3_	2.2	Phorbatopsin E (CECJNLRWMYCRSS-UHFFFAOYSA-N)	1
5.88		173.0708	146.0598			X		218.0920	C_11_H_12_N_3_O_2_	1.9	Methoxy phorbatopsin A (MDHOCGCTCYWXMY-TWGQIWQCSA-N)	2
6.07					X		X	342.1463	C_18_H_20_N_3_O_4_	−1.5	Calcaridine C (SGBQZSSTVLPIET-UHFFFAOYSA-N)	1
6.26					X		X	342.1450	C_18_H_20_N_3_O_4_	−0.2	Calcaridine C isomer (SGBQZSSTVLPIET-UHFFFAOYSA-N)	2
7.13			270.1488			X		340.1656	C_19_H_22_N_3_O_3_	−0.1	(AUMUDBPKOINNCL-UHFFFAOYSA-N)	3
6.31				146.0601		X		232.1080	C_12_H_14_N_3_O_2_	0.2	Leucettamine C (GWKCHEJMMQELNU-YFHOEESVSA-N)	2

## Data Availability

The MS/MS and NMR data presented in this study are openly available in Zenodo at https://doi.org/10.5281/zenodo.7152302. The obtained molecular network can be accessed at: https://gnps.ucsd.edu/ProteoSAFe/status.jsp?task=527472b15d4247dfad0534aa80f7ebfa.

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
