# Peer review of "An Aminopyrimidone and Aminoimidazoles Alkaloids from the Rodrigues Calcareous Marine Sponge Ernsta naturalis"

_marinedrugs, 2022, doi:10.3390/md20100637_

Round 1

Reviewer 1 Report

Dear authors!

The manuscript of paper by P-E. Camos et al. need to some little improves to become better. 

First of all, authors should add to the introduction data about molecular networks approach. 

Line 197: name of sponge Ernsta naturalis need be italicized.

Line 230: extra "the" word 

Author Response

The manuscript of paper by P-E. Camos et al. need to some little improves to become better. 

Thank you very much for your time and your comments.

First of all, authors should add to the introduction data about molecular networks approach.

An explanation about the molecular networks approach have been added to the introduction.

Line 197: name of sponge Ernsta naturalis need be italicized.

This mistake has been corrected

Line 230: extra "the" word 

This mistake has been corrected

Reviewer 2 Report

The paper reports the isolation of 8 new alkaloids, one aminopyridone and seven aminoimidazoles from a calcareus sponge belonging to a new genus. The new compounds are close analogs of  the large family of imidazole alkaloids from Leucetta and Chlathrina sponges, therefore the structural assignment is well defined. The configuration of the chiral center in 2, 3 and 4 was left unassigned, on the other hand it is not easy. I’d add a short explicative sentence on this aspect.

The use of the LC/ HRESIMS coupled to molecular network to analyze the metabolome of the sponge is interesting and it is useful to disclose minor components. In my opinion this analysis should always accompany the traditional work of isolation and characterization.

The editing of the experimental data is very accurate, especially in the experimental section.

As minor corrections:

In the introduction section three referencing don’t follow the journal style

In figure 2 and Table 2 phorbatopsin D was wrongly named phorbatopsin C.

The molecular formula usually refer to the neutral molecule rather to the observed pseudomolecular ion [M+H]+. Please correct throughout.

Page 5 line 161 Tautomerism is not an equilibrium between conformations! PLESE CORRECT.

Author Response

Thank you very much for your fell back and your comments. Please find our responses here.

Reviewer :

The paper reports the isolation of 8 new alkaloids, one aminopyridone and seven aminoimidazoles from a calcareus sponge belonging to a new genus. The new compounds are close analogs of the large family of imidazole alkaloids from Leucetta and Chlathrina sponges, therefore the structural assignment is well defined. The configuration of the chiral center in 2, 3 and 4 was left unassigned, on the other hand it is not easy. I’d add a short explicative sentence on this aspect.

Authors :

Calcaridine C.

If the NMR data of this compound is compared to those of calcaridine A and the epi-calcaridine, a significant difference is observed for the chemical shifts of protons H-6 i.e. :
     - 3.16/2.50 [1] or 3.13/2.47 [2] ppm for calcaridine A
     - 3.43 / 3.18 ppm for epi-calcaridine-A
     - 2.94/2.35 ppm for the Calcaridine-C, which is more consistent with the stereochemistry of Calcaridine-A (4R, 8S).

Calcaridine A is also a natural compound isolated from a calcareous sponge; the absolute configuration of calcaridine C should be the same : 4R, 8S. However the optical rotation couldn’t be determinant to confirm this hypothesis. Indeed, [a]D of the calcaridine A is +1.6° [2]., Therefore, with a value so close to 0, the comparison can’t be significant herein. So, due to this lack of certitude, we did not attribute the stereochemistry of the molecule in this publication.

Phorbatopsins D and E.

The stereochemistry of the carbon C-5 of phorbatopsins D and E can’t be determined by comparing their optical rotation to those of the known phorbatopsins B and C. No optical rotation was indeed mentioned in the literature for the latter. And as for these two molecules, any NOE correlation can be relevant due to the free rotation around the bond C-5 – C-6. The quantity and the purity of Phorbatopsins D and E prevent us to performe ECD spectra to determine the absolute configuration of these molecules.

|1] Koswatta, P. B.; Sivappa, R.; Dias, H. V. R.; Lovely, C. J. Total Synthesis of (±)-Calcaridine A and (±)-Epi-Calcaridine A. Org. Lett. 2008, 10 (21), 5055–5058. https://doi.org/10.1021/ol802018r.

[2] Edrada, R. A.; Stessman, C. C.; Crews, P. Uniquely Modified Imidazole Alkaloids from a Calcareous Leucetta Sponge. J. Nat. Prod. 2003, 66 (7), 939–942. https://doi.org/10.1021/np020503d.

[3] Nguyen, T. D.; Nguyen, X. C.; Longeon, A.; Keryhuel, A.; Le, M. H.; Kim, Y. H.; Chau, V. M.; Bourguet-Kondracki, M.-L. Antioxidant Benzylidene 2-Aminoimidazolones from the Mediterranean Sponge Phorbas Topsenti. Tetrahedron 2012, 68 (45), 9256–9259. https://doi.org/10.1016/j.tet.2012.08.074.

Reviewer :

The use of the LC/ HRESIMS coupled to molecular network to analyze the metabolome of the sponge is interesting and it is useful to disclose minor components. In my opinion this analysis should always accompany the traditional work of isolation and characterization.

The editing of the experimental data is very accurate, especially in the experimental section.

Authors :

Thank you very much for these comments.

Reviewer :

In the introduction section three referencing don’t follow the journal style

Authors :

Two references have been changed (12 and 13) but we are not sure about the third. We notice a little mistake in the reference 5, this is it ?

Reviewer :

In figure 2 and Table 2 phorbatopsin D was wrongly named phorbatopsin C.

Authors :

This mistake has been corrected

Reviewer :

The molecular formula usually refer to the neutral molecule rather to the observed pseudomolecular ion [M+H]+. Please correct throughout.

Authors :

The molecular formula has been changed for the neutral molecule for all molecules characterization. The pseudomolecular [M+H]+ have been kept when HRMS data are compared to exact masses.

Reviewer :

Page 5 line 161 Tautomerism is not an equilibrium between conformations! PLESE CORRECT.

Authors :

This mistake has been corrected.

Reviewer 3 Report

I recommend publish 

minor comments 

Perhaps its is best to define the stereochemistry in thymidine in the scheme unless you aren't sure about it

The legend in table 2 mentions phorb... D and E but the table has the column identities of phorb... C and D, this needs correcting 

Coupling constants are given as J=XX Hz I'm not sure what the journal style is but i thought you needed spaces like J = XX Hz

Apart from this, the paper was very interesting 

Author Response

Reviewer : I recommend publish 

Authors : Thank you very much for your feel back.

Reviewer : Perhaps its is best to define the stereochemistry in thymidine in the scheme unless you aren't sure about it.

Authors : Stereochemistry of thymidine has been added.

Reviewer : The legend in table 2 mentions phorb... D and E but the table has the column identities of phorb... C and D, this needs correcting 

Authors : This mistake has been corrected.

Reviewer : Coupling constants are given as J=XX Hz I'm not sure what the journal style is but i thought you needed spaces like J = XX Hz

Authors : The typography has been changed for the correct one.